# Dexmedetomidine use in pediatric strabismus surgery: A systematic review and meta-analysis

**Fu-Wei Chiang**[1,2☯], **Jin-Lin Chang**[1,2☯], **Shih-Chang Hsu**[1,2], **Kuo-Yuan Hsu**[1,2], **Karen Chia-Wen Chu**[1,2], **Chun-Jen Huang**[3], **Chyi-Huey Bai**[4], **Chiehfeng Chen**[5,6,7], **Chin-Wang Hsu**[1,2], **Yuan-Pin Hsu**[1,2]*

**1** Emergency Department, Wan Fang Hospital, Taipei Medical University, Taipei, Taiwan, **2** Department of Emergency, School of Medicine, College of Medicine, Taipei Medical University, Taipei, Taiwan, **3** Department of Anesthesiology, Wan Fang Hospital, Taipei Medical University, Taipei, Taiwan, **4** Department of Public Health, School of Medicine, College of Medicine, Taipei Medical University, Taipei, Taiwan, **5** Evidence-Based Medicine Center, Wan Fang Hospital, Taipei Medical University, Taipei, Taiwan, **6** Cochrane Taiwan, Taipei Medical University, Taipei, Taiwan, **7** Division of Plastic Surgery, Department of Surgery, Wan Fang Hospital, Taipei Medical University, Taipei, Taiwan

☯ These authors contributed equally to this work.
* koakoahsu@gmail.com

**Data Availability Statement:** All relevant data are within the manuscript and its Supporting Information files.

**Funding:** The authors, Yuan-Pin, Hsu and Jin-Lin Chang, are very grateful for financial support from

## Abstract

### Background

Common complications of pediatric strabismus surgery, including emergence agitation (EA), postoperative nausea and vomiting (PONV), and postoperative pain, may be prevented using dexmedetomidine, which is an anxiolytic and analgesic. This systematic review and meta-analysis assessed the effects of dexmedetomidine in patients who had undergone pediatric strabismus surgery.

### Method

Five databases were searched for randomized controlled trials published from database inception to April 2020 that compared dexmedetomidine use with placebo or active comparator use and evaluated EA, PONV, or postoperative pain incidence (main outcomes) in patients who had undergone pediatric strabismus surgery. Oculocardiac reflex (OCR) incidence and postanesthesia care unit (PACU) stay duration were considered as safety outcomes. All meta-analyses were performed using a random-effects model.

### Results

In the nine studies meeting our inclusion criteria, compared with placebo use, dexmedetomidine use reduced EA incidence [risk ratio (RR): 0.39; 95% confidence interval (CI): 0.25–0.62, $I^2 = 66\%$], severe EA incidence (RR: 0.27, 95% CI: 0.17–0.43, $I^2 = 0\%$), PONV incidence (RR: 0.33, 95% CI: 0.21–0.54, $I^2 = 0\%$), analgesia requirement (RR: 0.38, 95% CI: 0.25–0.57, $I^2 = 0\%$), and pain scores (standardized mean difference: −1.02, 95% CI: −1.44 to −0.61, $I^2 = 75\%$). Dexmedetomidine also led to lower EA incidence in the sevoflurane

project no. 109-wf-eva-12 of Wan Fang Hospital, Taipei Medical University, Taipei, Taiwan.

**Competing interests:** The authors declare no conflict of interests.

group than in the desflurane group (RR: 0.26 for sevoflurane vs. 0.45 for desflurane). Continuous dexmedetomidine infusion (RR: 0.19) led to better EA incidence reduction than did bolus dexmedetomidine infusion at the end of surgery (RR: 0.26) or during the peri-induction period (RR: 0.36). Compared with placebo use, dexmedetomidine use reduced OCR incidence (RR: 0.63; $I^2$ = 40%). No significant between-group differences were noted for PACU stay duration.

## Conclusion

In patients who have undergone pediatric strabismus surgery, dexmedetomidine use may alleviate EA, PONV, and postoperative pain and reduce OCR incidence. Moreover, dexmedetomidine use does not affect the PACU stay duration.

## Introduction

Pediatric strabismus surgery is one of the most common ophthalmic procedures performed under general anesthesia. However, after strabismus surgery, many pediatric patients present with emergence agitation (EA), postoperative nausea and vomiting (PONV), and postoperative pain [1], whose incidence rates are as high as 40%–86% [1], 37%–80% [2], and 65% [3], respectively. Moreover, EA can increase the risk of inadvertent removal of intravenous catheters and self-harm; PONV can have adverse consequences such as dehydration, electrolyte imbalance, delayed hospital discharge, and unplanned hospital admission [4]; and postoperative pain can lead to decreased oral intake and dehydration as well as delayed discharge from the hospital [5]. EA, PONV, and postoperative pain after strabismus surgery may thus increase stress among medical staff as well as caregivers.

Dexmedetomidine is a highly selective α 2-adrenoreceptor agonist with sedative, analgesic, and anxiolytic properties. It has been widely used in clinical practice [6]. Dexmedetomidine can be used to prevent or treat delirium in the intensive care unit [7] and prevent EA in adult patients who have undergone cardiac and noncardiac surgery [8]. Cho et al. reported that the perioperative administration of dexmedetomidine can provide pain and agitation relief without side effects in children undergoing adenotonsillectomy [9]. In a meta-analysis, dexmedetomidine premedication was found to reduce PONV incidence in children undergoing different types of surgery [10]. However, only one of the 13 included studies in the aforementioned meta-analysis focused on pediatric strabismus surgery [10]. Therefore, whether dexmedetomidine can reduce EA, PONV, and postoperative pain incidence in pediatric patients undergoing strabismus surgery remains unclear. Moreover, relevant studies, which have only used small sample populations, have reported conflicting results [11–19]. Thus, meta-analyses evaluating the safety and efficacy of dexmedetomidine in pediatric strabismus surgery are lacking.

In this research, we conducted a systematic review and meta-analysis of randomized controlled trials (RCTs) evaluating the protective efficacy and safety of dexmedetomidine in pediatric patients undergoing strabismus surgery.

## Methods

### Search strategy and study eligibility criteria

We searched the Cochrane Library, EMBASE, PubMed, Web of Sciences, and Scopus databases to identify eligible research published from database inception until April 2020. The

keywords used included "dexmedetomidine," "$\alpha_2$ agonist," "children," "pediatric," "eye surgery," "strabismus," and "ophthalmic surgery." These terms and their combinations were also searched as text words. After eligible studies were selected, the references in these studies were reviewed manually to identify additional relevant studies. We also searched the ClinicalTrials.gov registry (http://clinicaltrials.gov/) for any unpublished relevant studies. No limitation filter was used. The search strategy for each database is detailed in S1 Table.

We only included human RCTs on strabismus surgery that compared dexmedetomidine use with placebo or active comparator use (any administration route or dose) in patients aged <18 years. All reviews, cohort studies, case series, and case reports were excluded. Subsequently, two independent reviewers (FWC and KYH) removed duplicate references, screened the titles and abstracts of the remaining articles, and then examined the full text of the articles to identify eligible RCTs. All disagreements were resolved after discussion with a third author (YPH) to achieve consensus.

Our main outcomes of interest were the incidences of EA, PONV, and postoperative pain (including number of patients requiring rescue analgesia and pain scores). Moreover, our safety outcomes included postanesthesia care unit (PACU) stay duration and oculocardiac reflex (OCR) incidence.

## Data extraction

Two reviewers (KCWC and YPH) independently abstracted the following information from the included studies: first author, publication year, time of patient recruitment, inclusion criteria, sample sizes, baseline characteristics, regimens of each comparison, intervention or control timing and duration, main anesthetic drug, and main and safety outcome data. We followed the recommendations of the Cochrane Handbook for Systematic Reviews of Interventions (https://handbook-5-1.cochrane.org/) if the included study used a multiple-arm design. For multiple arms in a study, each pairwise comparison was included separately and shared controlled groups were divided nearly evenly among the comparisons.

## Methodological quality of the included studies

Two authors (SCH and KCWC) independently used the Cochrane Risk of Bias 2.0 tool to evaluate the methodological quality of the included studies [20]. This tool examines the following biases: selection bias due to the inadequate generation of a randomized sequence or the concealment of allocations before the assignment, performance bias due to knowledge of the allocated interventions by participants and personnel, detection bias due to knowledge of the allocated interventions by outcome assessors, reporting bias due to selective outcome reporting, and other biases. Accordingly, we summarized the results in a risk of bias graph. If the judgment of the aforementioned two authors (SCH and KCWC) was conflicting, a third senior investigator (CC) was consulted for the final judgment.

## Statistical analysis

For dichotomous outcomes, we calculated the risk ratio (RRs) with their corresponding 95% confidence intervals (CIs). Continuous data are presented using the mean difference (MD) with its corresponding 95% CI. When a study did not report mean and variance values, we estimated these values on the basis of the reported sample sizes, medians, and interquartile ranges [21]. If all studies assessed a continuous outcome variable by using varied approaches, we calculated the standardized MD (SMD) with its corresponding 95% CIs. We interpreted the magnitude of the SMD as follows: SMD = 0.2, small; SMD = 0.5, medium; and SMD = 0.8,

large [22]. Moreover, we used the DerSimonian and Laird random-effects model to synthesize the results for the outcome of interest.

The heterogeneity among studies in each analysis was measured using the $I^2$ statistic and $\chi^2$ test. If substantial heterogeneity ($I^2 > 50$) was identified, we explored possible underlying causes by performing prespecified subgroup analyses. Subgroups were obtained according to the main anesthetics used (i.e., desflurane and sevoflurane), intervention timing and duration (given as continuous infusion during surgery or as bolus infusion in the peri-induction period or at the end of surgery), and measurement method. Sensitivity analysis was performed to explore heterogeneity for the primary outcome by using the one-study-out method and by restricting RCTs at a low risk of bias. Moreover, funnel plots were used for assessing publication bias and testing the symmetry of the funnel plots by using Egger's test [23].

We performed all meta-analyses using Review Manager (version 5.3; Cochrane Collaboration, Copenhagen, Denmark) or STATA (version 14.0; Stata Corp, College Station, TX, USA). A two-sided $P$ of <0.05 was considered to indicate statistical significance.

**Trial sequential analysis.** A trial sequential analysis (TSA) was performed to reduce the risk of random errors, increase the robustness of the meta-analyses, and determine whether the current sample size was sufficient [24, 25]. TSA monitoring boundaries for the meta-analysis and the required information size (RIS) were quantified, and adjusted CIs were calculated. The RIS indicated a target sample size considering the heterogeneity of the data. The risk of a type 1 error was set to 5% with a power of 90%. If the cumulative z-curve crosses the trial sequential monitoring boundary, a sufficient level of evidence has been reached, and no further trials are needed. If the z-curve does not cross the boundary and the required information size has not been reached, there is insufficient evidence to reach a conclusion. The TSA program vers. 0.9.5.5 beta (Copenhagen Trial Unit, Copenhagen, Denmark) was used for the TSAs.

## Protocol registration

Our meta-analysis was conducted in accordance with the Preferred Reporting Items for Systematic Review and Meta-Analyses (PRISMA) guidelines (S1 Checklist) [20] and registered in the PROSPERO database (number: CRD42018091450).

## Results

### Search results

Fig 1 displays our study screening and selection strategy. A comprehensive search of the Cochrane Library, EMBASE, PubMed, Scopus, and Web of Sciences databases in conjunction with manual search produced 345 records. After removing 138 duplicate records and excluding 185 citations by screening their titles and abstracts, 22 full-text articles were examined in detail. We then excluded 13 articles because they were commentary articles not RCT designs or studies not involving pediatric strabismus surgery. Finally, nine eligible studies were included for further qualitative and quantitative analyses [11–19]. The characteristics of the included studies are summarized in Table 1.

### Risk of bias assessment

Fig 2 illustrates the risk of bias assessment results. Most of the identified studies were rated to have a low risk of bias. We rated two studies as having high risk of bias: one study was conducted without blinding of allocation and concealment [14] and another study had a high risk

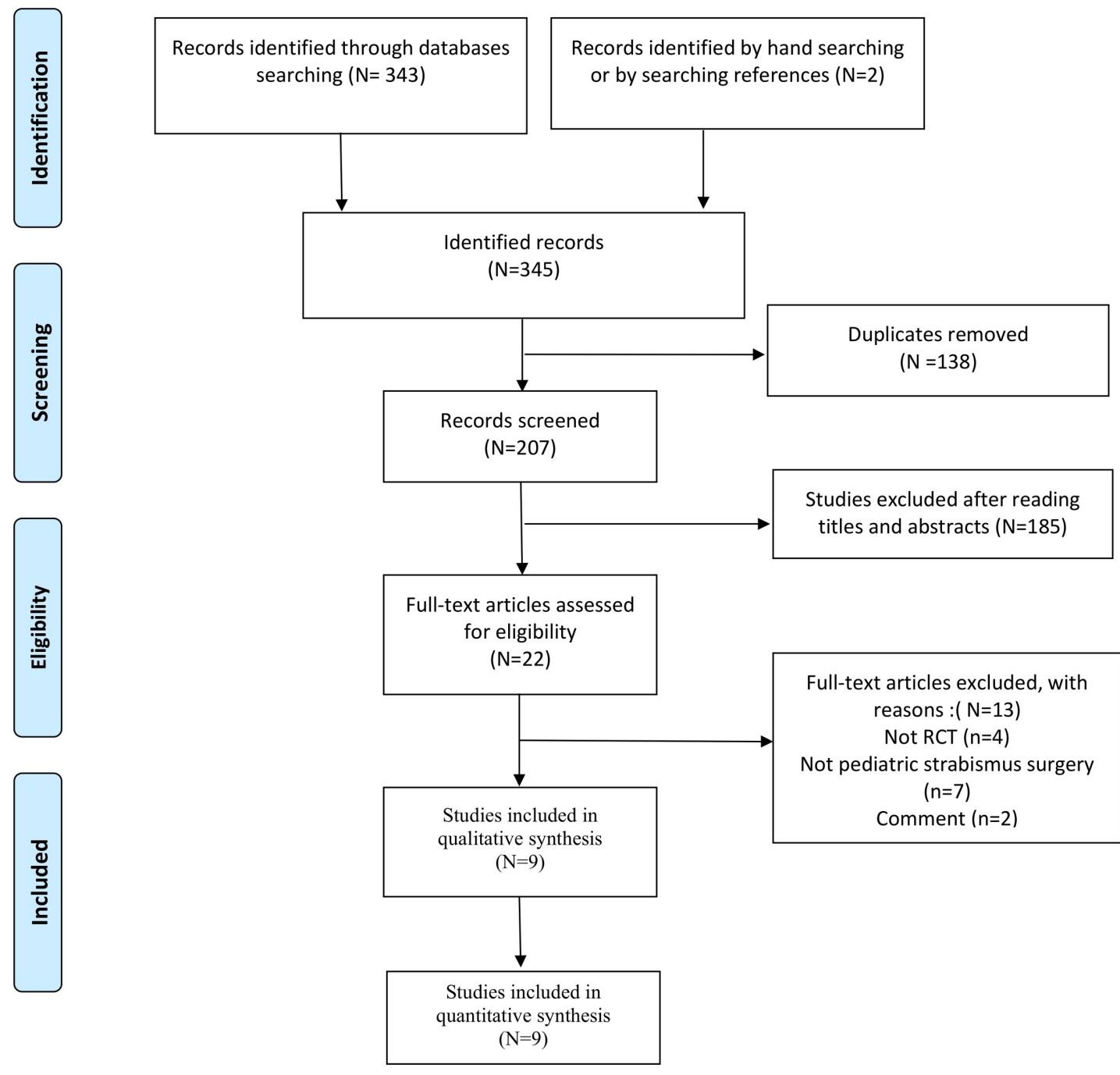

**Fig 1. Study screening and selection strategy.**

in the outcome assessment domain [16]. Moreover, three studies were rated as having an unclear risk of other biases because the prespecified sample size was not calculated [11, 16, 18].

## Pooled results of the included studies

**Dexmedetomidine versus placebo (saline).** *EA incidence*. Eight studies [11–16, 18, 19] included 11 pairwise comparisons evaluating EA incidence. Our meta-analysis with 642

**Table 1. Characteristics of the included studies.**

| First author, year, and country | Time of recruitment | Inclusion criteria | Sample size | Age, years | Sex (M/F) | Bodyweight, kg | Intervention | Time and duration of intervention or control | Main anesthetic used |
|---|---|---|---|---|---|---|---|---|---|
| Mizrak 2011 [18], Turkey | September to November 2009 | Age: 4–11 years, Strabismus, ASA I | 30 | 8.5 (2.6) | 15/15 | 28.6 (10.7) | Dex: IV, 0.5 µg/kg, | Start 10 min before induction | Ketamine at a rate of 1–2 mg/kg/h |
| | | | 30 | 8.6 (2.8) | 13/17 | 29.5 (7.8) | C: normal saline | | |
| Chen 2013 [13], China | September 2010 to January 2011 | Age: 2–7 years, Strabismus, ASA I–II | 27 | 4.1 (1.1) | 17/10 | 17.3 (4.2) | Dex: IV, 1.0 µg/kg, maintain: 0.25 mL/kg/h | Start over 1 min after induction and maintain until the end of surgery | 3%–4% sevoflurane |
| | | | 27 | 4.2 (1.2) | 18/9 | 17.7 (4.1) | K: IV, 1 mg/kg | | |
| | | | 24 | 4.3 (1.1) | 15/9 | 18.0 (3.6) | C: normal saline, | | |
| Kim 2014 [15], South Korea | September 2011 to March 2012 | Age: 1–5 years, Strabismus, ASA I–II | 47 | 4.3 (1.4) | 18/29 | 18.8 (5.4) | Dex: IV, 0.2 µg/kg/h | Start after induction; continuous fusion until the end of surgery | Desflurane |
| | | | 47 | 4.3 (1.0) | 26/21 | 18.3 (3.7) | C: normal saline, | | |
| Abdelaziz 2016 [11], Saudi Arabia | September 2013 to April 2015 | Age: 1–7 years, Strabismus, ASA I–II | 33 | 2.7 (1.5) | 17/16 | 12 (3.9) | Dex: IN,1µg/kg | Start before induction | Sevoflurane and 50% N₂O in oxygen |
| | | | 33 | 2.5 (1.2) | 17/16 | 11.8 (3.7) | Mi: IN, 0.1 mg/kg | | |
| | | | 32 | 2.8 (1.7) | 18/14 | 11.4 (3.3) | C: normal saline | | |
| Song 2016 [19], South Korea | February 2013 to February 2014 | Age: 2–6 years, Strabismus, ASA I | 28 | 4.6 (1.3) | 16/12 | 19.7 (5.1) | Dex: IV, 1.0 µg/kg | Start 10 min after induction | 8–10% desflurane and 60% N₂O. |
| | | | 28 | 4.5 (1.3) | 10/18 | 19.1 (4.8) | Dex: IV, 0.5 µg/kg | | |
| | | | 28 | 4.3 (1.7) | 14/14 | 18.4 (4.5) | Dex: IV, 0.25 µg/kg | | |
| | | | 28 | 3.8 (1.5) | 14/14 | 18.1 (4.2) | C: normal saline | | |
| Abdel-Rahman 2018 [12], Egypt | March to December 2016 | Age: 3–8 years, Strabismus, ASA I–II | 30 | 4.5 (1.0) | 22/8 | 16.7 (1.7) | Dex: IV,0.5 µg/kg | Start 10 min at the end of the surgery with the closure of conjunctiva | 2%–4% sevoflurane |
| | | | 30 | 4.4 (1.2) | 19/11 | 16.5 (2.6) | Dex: IV,0.25 µg/kg | | |
| | | | 30 | 4.6 (1.2) | 21/9 | 17. 0(2.6) | C: normal saline | | |
| Lee 2018 [16], China | Not reported | Age: 4–8 years, Strabismus, ASA I–II | 60 | 6.2 (2.2) | 31/29 | 22 (8) | Dex: IV,0.6 µg/kg | Start 15 min before induction | 1%–2% sevoflurane + sufentanil 0.1–0.3 µg/kg/h |
| | | | 60 | 6.2 (1.8) | 35/25 | 21 (9) | C: normal saline | | |
| Hamawy 2019 [14], Egypt | January 2015 to June 2015 | Age: 2–10 years, Strabismus, ASA I–II | 25 | 5.3 (1.6) | 15/10 | 19.8 (5) | Dex, IN, 1 µg/kg | Start 30 min before induction | Sevoflurane and 50% air in oxygen |
| | | | 25 | 5.76 (2) | 16/9 | 20. 5(4.3) | C: normal saline | | |
| Li 2020 [17], China | December 2018 to March 2019 | Age: 6–10 years, Strabismus, ASA I–II | 40 | 8.3 (1.1) | 20/20 | 16.6 (3.9) | Dex: IV, 0.3 µg/kg | Drug was administered IV every 10 min | Sevoflurane and 50% air in oxygen with a constant fresh gas flow of 2 L/min |
| | | | 41 | 8.3 (1.1) | 19/22 | 16.2 (2.8) | Dex: IV, 0.5 µg/kg | | |
| | | | 41 | 8.2 (1.3) | 24/17 | 17.1 (3.0) | C: normal saline | | |

ASA, American Society of Anesthesiologists; Dex, dexmedetomidine; F, Female; M, male; Mi, midazolam; C, control; K, ketamine; IN, intranasally; IV, intravenously

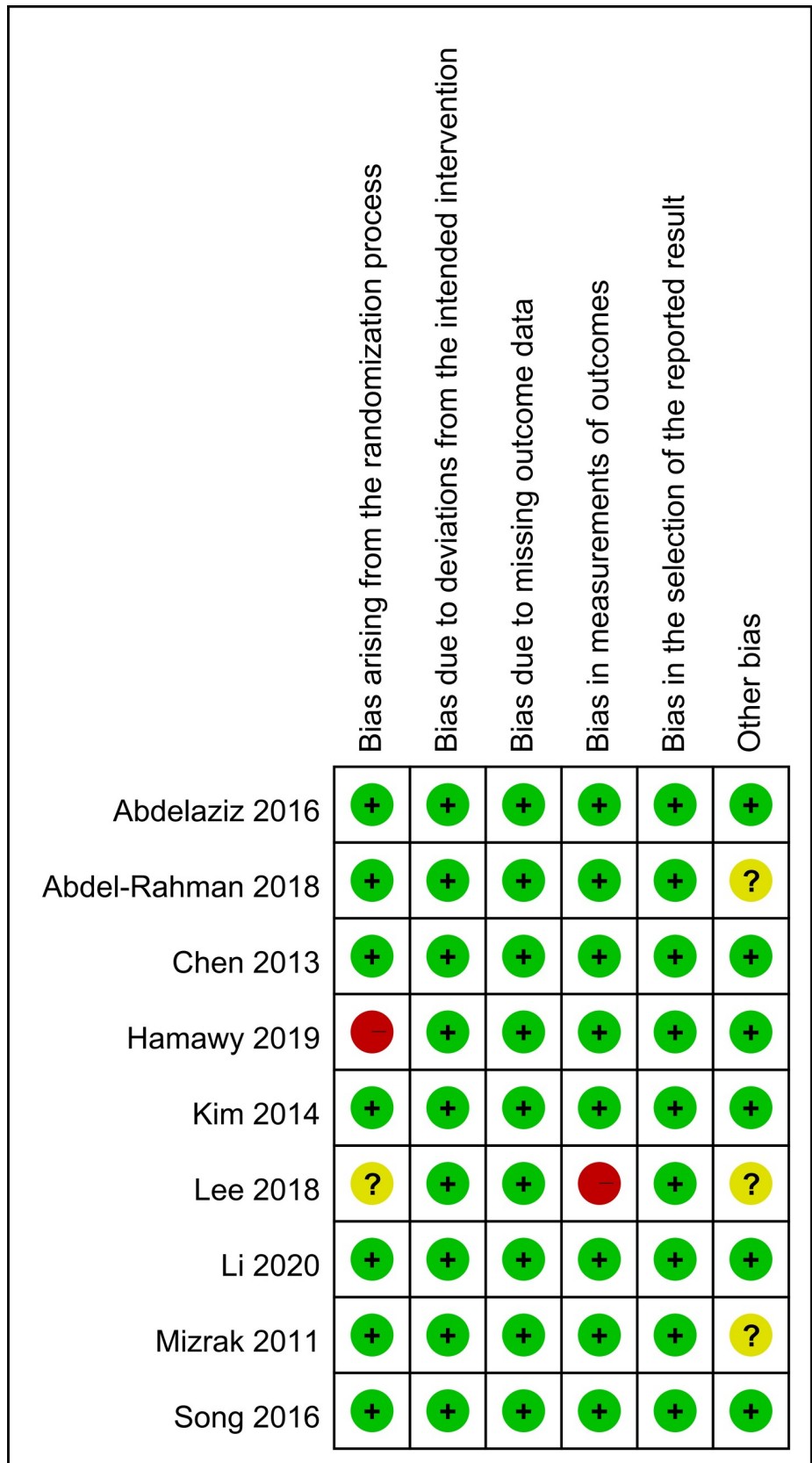

**Fig 2. Summary of risk of bias assessment.**

A.

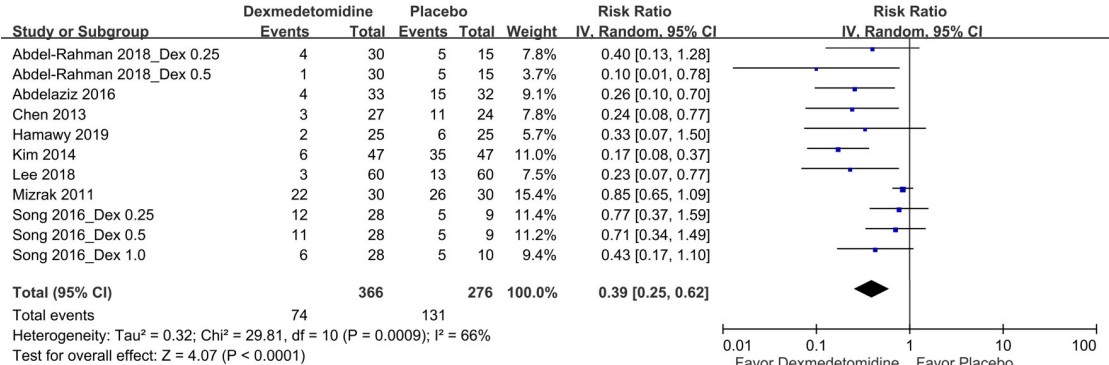

B.

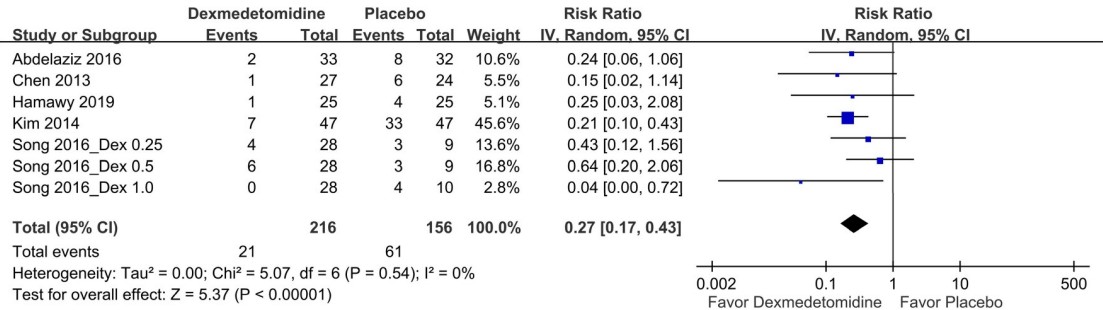

**Fig 3.** Forest plot for (A) EA and (B) severe EA incidence. Dex, dexmedetomidine.

patients in all the included pairwise comparisons indicated that dexmedetomidine use significantly reduced EA incidence compared with placebo use (Fig 3A, pooled RR: 0.39, 95% CI: 0.25–0.62). Moreover, the heterogeneity for EA incidence in the eight studies was substantial ($I^2$ = 66%). Next, we performed subgroup analysis according to the main anesthetics used (Table 2). The result revealed that dexmedetomidine use significantly led to a lower EA incidence than did placebo use when patients received sevoflurane (pooled RR: 0.26, 95% CI: 0.16–0.44) or desflurane (pooled RR: 0.45, 95% CI: 0.22–0.91) (S1 Fig). The sevoflurane subgroup demonstrated no heterogeneity ($I^2$ = 0%). Notably, the effect size was significantly larger in the sevoflurane subgroup than in the desflurane (test of subgroup difference: $P < 0.05$). In addition, only one study used ketamine as the main anesthetic, which precluding the meta-analysis in this subgroup. In this study, the authors reported that dexmedetomidine use did not lead to a lower EA incidence compared to placebo use. Additional subgroup analyses were performed according to the timing and duration of dexmedetomidine administration (Table 2). Compared with placebo use, dexmedetomidine administered as bolus infusion in the peri-induction period (pooled RR: 0.55, 95% CI: 0.37–0.83) or at the end of surgery (pooled RR: 0.26, 95% CI: 0.08–0.92) or as a continuous infusion (pooled RR: 0.19, 95% CI: 0.10–0.36) significantly reduced EA incidence (S2 Fig). The effect sizes were significantly larger in patients receiving bolus infusion at the end of surgery or continuous infusion than in those receiving bolus infusion in the peri-induction period (test of subgroup difference: $P < 0.05$). We performed further subgroup analyses of a different measurement method for EA (Table 2).

**Table 2. Subgroup analysis results.**

| Category | Subgroups | No. of pairwise comparison | No. of patients | RR [95% CI] | P | Group heterogeneity $I^2$ | Subgroup difference P |
|---|---|---|---|---|---|---|---|
| Outcome: EA incidence (dexmedetomidine vs. placebo) | | | | | | | |
| Main anesthetics | Sevoflurane | 6 | 376 | 0.26 [0.16, 0.44] | <0.05* | NA | <0.05* |
| | Desflurane | 4 | 206 | 0.45 [0.22, 0.91] | <0.05* | 69 | |
| Timing and duration of dexmedetomidine administration | Bolus, peri-induction period | 7 | 407 | 0.55 [0.37, 0.83] | <0.05* | 47 | <0.05* |
| | Continuous infusion | 2 | 145 | 0.19 [0.10, 0.36] | <0.05* | 0 | |
| | Bolus, at the end of surgery | 2 | 90 | 0.26 [0.08, 0.92] | <0.05* | 25 | |
| EA incidence measurement method | PAED scale | 7 | 436 | 0.34 [0.18, 0.66] | <0.05* | 66 | 0.58 |
| | 4-point EA scale | 4 | 206 | 0.39 [0.22, 0.91] | <0.05* | 69 | |

Footnote: EA, emergence agitation; PAED, Pediatric Anesthesia Emergence Delirium; RR, risk ratio; NA, not applicable

*, statistically significant.

According to the results of the Pediatric Anesthesia Emergence Delirium (PAED) scale (pooled RR: 0.34, 95% CI: 0.18–0.66) and 4-point EA scale (pooled RR: 0.45, 95% CI: 0.22–0.91) for assessing EA incidence, dexmedetomidine led to significantly lower EA incidence than did placebo (S3 Fig). No significant subgroup differences were noted in the aforementioned analysis. Moreover, the results of the sensitivity analysis performed using the one-study-out method indicated that the aforementioned finding was robust (S4 Fig). Another sensitivity analysis by restricting RCTs at a low risk of bias had no effect on the result (S5 Fig).

Five studies [11, 13–15, 19] included seven pairwise comparisons that evaluated severe EA incidence. A score of 4 was defined to indicate severe EA in the two studies [15, 19] that used a 4-point EA scale, whereas a score of ≥15 was used to define severe EA in the three studies [11, 13, 14] that used the PAED scale. The pooled results depicted in Fig 3B revealed that dexmedetomidine significantly reduced severe EA incidence (n = 372; pooled RR: 0.27, 95% CI: 0.17–0.43). No heterogeneity was noted for severe EA incidence in the aforementioned five studies ($I^2 = 0\%$)

*PONV incidence.* Six studies [11, 13, 14, 16, 17, 19] employed nine pairwise comparisons to examine PONV incidence. Our meta-analysis with 520 patients in all the included pairwise comparisons indicated that dexmedetomidine use led to significantly lower PONV incidence than did placebo use (Fig 4, Pooled RR: 0.33, 95% CI: 0.21–0.54). No heterogeneity was noted for PONV incidence in the six studies ($I^2 = 0\%$).

*Postoperative pain.* Four studies [11, 15, 18, 19] that involved six pairwise comparisons reported that patients required rescue analgesia. Moreover, 194 patients received dexmedetomidine and 137 received the placebo. The pooled results indicated that the number of patients requiring rescue analgesia was relatively lower in the dexmedetomidine group than in the placebo group (pooled RR: 0.38, 95% CI: 0.25–0.57; Fig 5A). The aforementioned four studies demonstrated no heterogeneity for the number of patients requiring rescue analgesia ($I^2 = 0\%$).

Six studies [11–15, 19] that involved nine pairwise comparisons evaluated postoperative pain scores. Regarding the tool used for pain severity measurement, three studies [11, 12, 19] used

the Face, Legs, Activity, Cry, Consolability scale, two [13, 14] used the Children's Hospital of Eastern Ontario Pain Scale, and one [15] used the objective pain scale. The meta-analysis of the pain scores revealed that dexmedetomidine use led to significantly lower pain score than did placebo use (n = 462; pooled SMD: −1.02, 95% CI: −1.44 to −0.61; Fig 5B). Substantial heterogeneity was noted for postoperative pain scores in the aforementioned six studies ($I^2$ = 75%).

*Safety outcomes*. Fig 6 illustrates the effects of dexmedetomidine use on OCR incidence. Five studies [11, 13, 17–19] that involved eight pairwise comparisons and comprised 410 patients reported on OCR incidence. The pooled results indicated that compared with placebo use, dexmedetomidine use significantly reduced OCR incidence, with a pooled RR of 0.63, a wide 95% CI of 0.41–0.97, and an $I^2$ value of 40%. Moreover, seven studies [11–15, 17, 19] that involved 11 pairwise comparisons and comprised 584 patients evaluated PACU stay duration. In general, PACU stay duration did not differ between the dexmedetomidine and placebo groups (Fig 7).

**Dexmedetomidine versus active comparators.** Because only two studies [11, 13] used an active comparator as the control (one used ketamine [13] and another used midazolam [13]), we did not pool the results. Chen et al. [13] reported EA incidence in 11% of patients using dexmedetomidine and 22% of patients using ketamine; however, the difference in EA incidence between the two groups was nonsignificant. PONV incidence was significantly lower in the dexmedetomidine group than in the ketamine group (PONV risk: 15% for dexmedetomidine vs. 44% for ketamine, *P* = 0.02). The between-group differences in the pain scores were nonsignificant [13]. Abdelaziz et al. [11] reported EA incidence was significantly lower in the dexmedetomidine group than in the midazolam group (EA risk: 12% for dexmedetomidine vs. 21% for midazolam). Moreover, PONV incidence was significantly lower in the dexmedetomidine group than in the midazolam group (PONV risk: 15% for dexmedetomidine vs. 21% for midazolam) [11]. However, according to the data provided by Abdelaziz et al. [11], the aforementioned significant findings were subject to statistical errors.

**Risk of bias across studies.** The funnel plots for EA incidence exhibited a skewed or asymmetrical shape (S6 Fig; Egger's test: *P* < 0.05). By contrast, the funnel plots for severe EA incidence, PONV incidence, number of patients requiring rescue analgesia, pain scores, OCR incidence, and PACU stay duration did not exhibit a skewed or asymmetrical shape (S7–S12 Figs, respectively; Egger's test: *P* = 0.94, 0.12, 0.75, 0.16, 0.78, and 0.57, respectively).

**Trial sequential analysis.** The cumulative Z-score crossed the conventional meta-analysis significance boundary as well as the trial sequential monitoring boundary for benefit indicating a significant effect on the outcome of EA incidence (S13 Fig), severe EA incidence (S14 Fig), PONV incidence (S15 Fig), patients requiring rescue analgesia (S16 Fig) and postoperative pain scores (S17 Fig) of dexmedetomidine compared to control; These cannot be ascribed to random error. This indicated that further studies are not likely to alter the conclusion. In the TSA for OCR incidence, the Z-curve crossed conventional meta-analysis significance boundary but not the TSA monitoring boundary (S18 Fig). The accrued information size (n = 410) did not reach the RIS (n = 478), indicating that the effect may change when new evidence accumulates. In the TSA for PACU stay duration, the Z-curve did not cross the TSA monitoring boundary or the futility boundary (S19 Fig). The accrued information size (n = 584) did not reach the RIS (n = 8482), denoting that there is insufficient evidence to reach a conclusion and additional trials may be required.

## Discussion

To the best of our knowledge, this research is the first systematic review and meta-analysis of RCTs assessing the effects of dexmedetomidine on EA, PONV, and postoperative pain

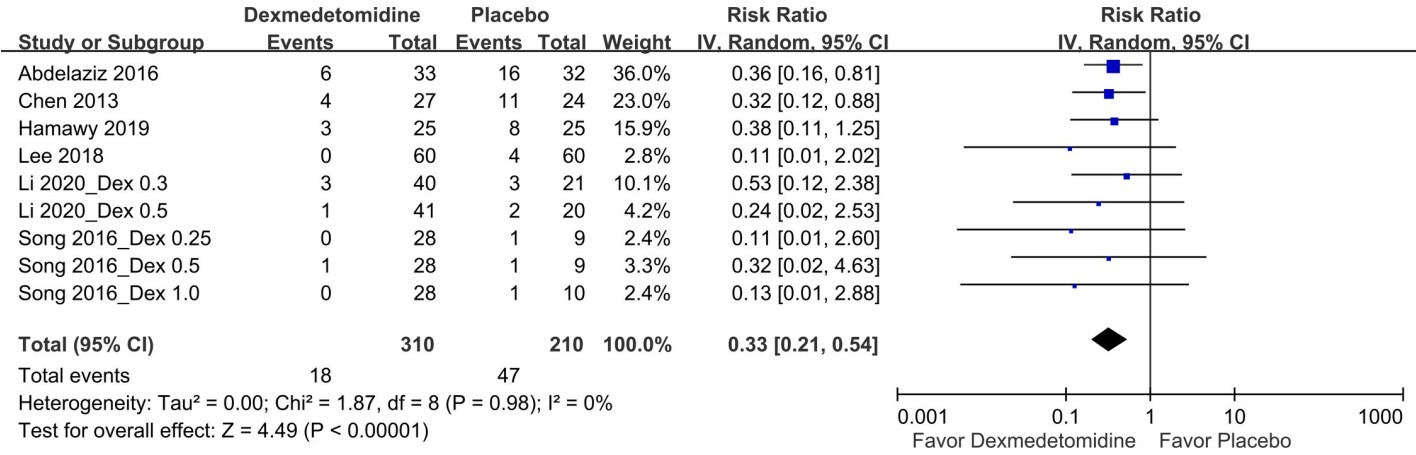

**Fig 4. Forest plot for PONV incidence.** Dex, dexmedetomidine.

incidence for patients who have undergone pediatric strabismus surgery. Nine studies were included in this meta-analysis after a comprehensive search of multiple electronic databases. Our final results indicated that dexmedetomidine use in pediatric strabismus surgery considerably reduced EA and severe EA incidence. Similarly, it reduced PONV incidence, pain scores, and number of patients requiring analgesia. Compared with saline use, dexmedetomidine use was associated with lower OCR occurrence. Moreover, dexmedetomidine use did not extend PACU stay duration.

A.

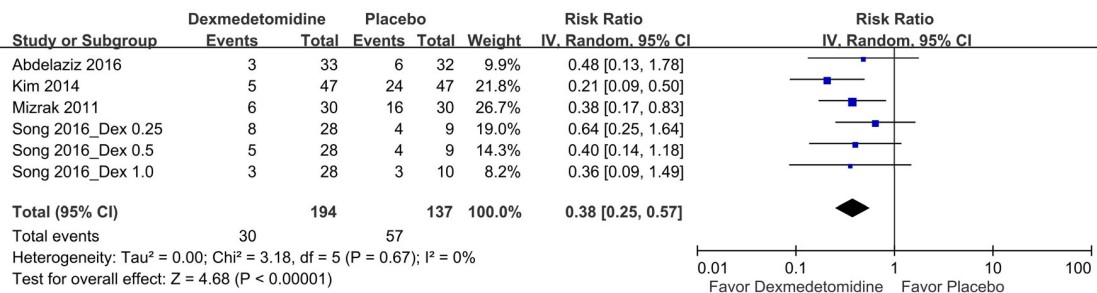

B.

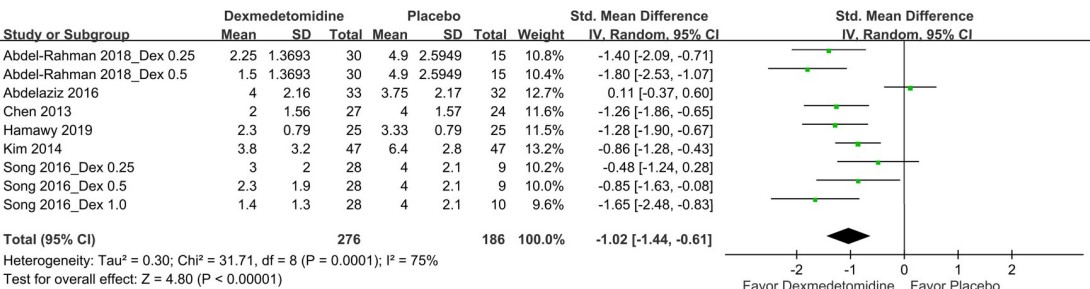

**Fig 5.** Forest plot for postoperative pain: (A) patient requiring rescue analgesia and (B) pain scores. Dex, dexmedetomidine.

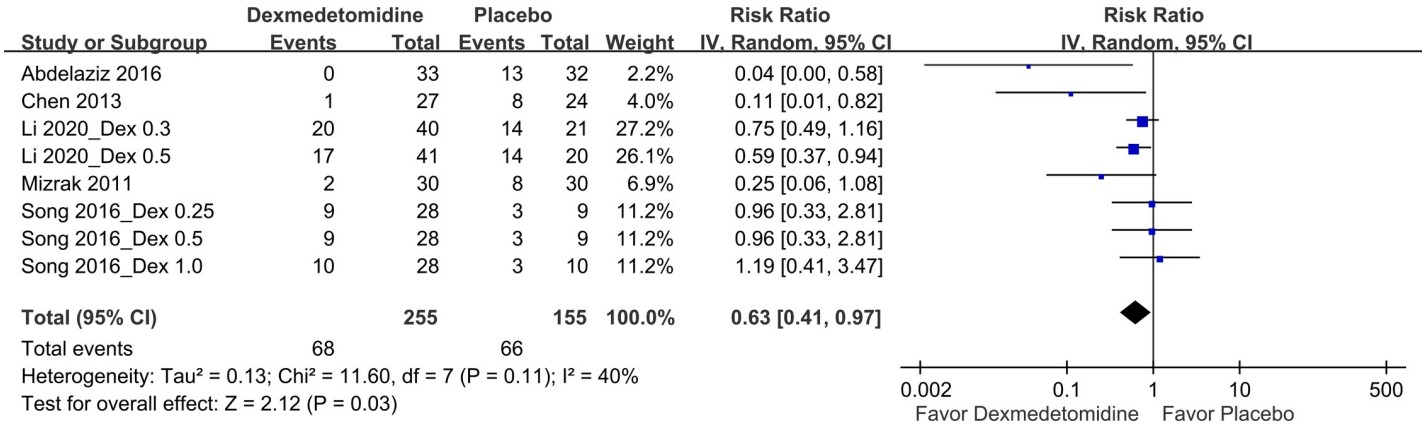

**Fig 6. Forest plot for OCR incidence.** Dex, dexmedetomidine.

EA is a common adverse postoperative complication in children, particularly preschoolers (aged 5–7 years) [26] and children undergoing strabismus surgery [1]. The possible risk factors of EA include rapid emergence from anesthesia, use of short-acting volatile anesthetic agents, postoperative pain, age, and surgery type [27]. However, the EA incidence in children, even within particular subgroups of pediatric patients (e.g., patients who have undergone pediatric strabismus surgery), remains under debate [27]. Duan et al. [8] reported that dexmedetomidine use reduced EA incidence in the entire adult surgical population. Moreover, Ni et al. [28] indicated that intravenous dexmedetomidine significantly reduced EA incidence in children undergoing various types of surgery; however, because the authors did not perform surgery-type-based subgroup analysis, the generalizability of their results was limited to only a specific population. Cho et al. [9] demonstrated that in children undergoing adenotonsillectomy, perioperative dexmedetomidine administration was associated with EA incidence reduction. Therefore, we suggest that in a specific high-risk EA population, such as pediatric patients undergoing strabismus surgery, clinicians should select dexmedetomidine to reduce EA occurrence.

During recovery from general anesthesia, inhalational anesthetic use in children can also frequently lead to EA. The use of sevoflurane and desflurane, which are short-acting volatile anesthetic agents, is an EA risk factor. Both these agents are commonly used in pediatric

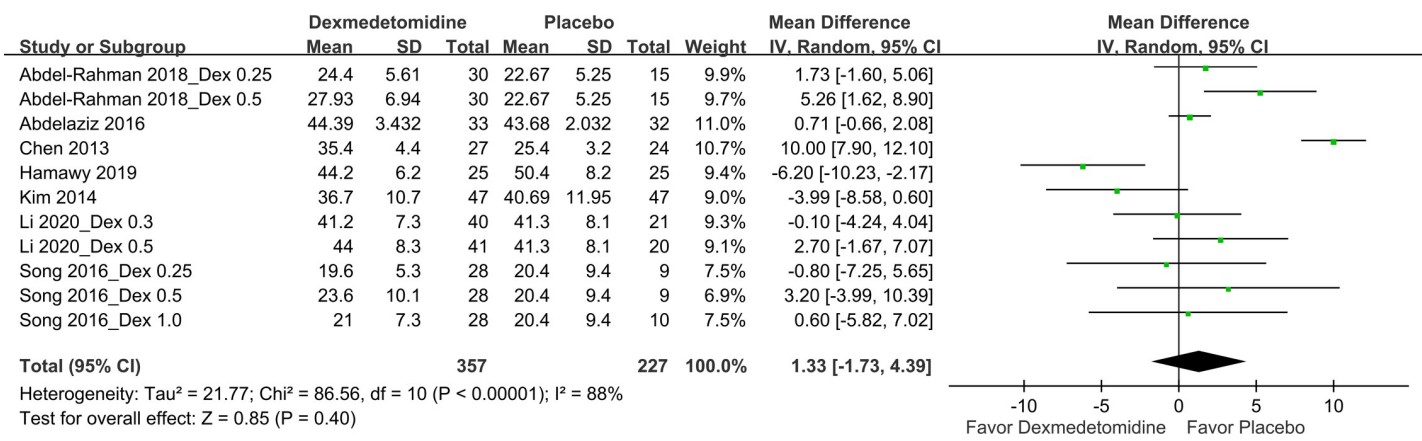

**Fig 7. Forest plot for PACU stay duration.** Dex, dexmedetomidine.

anesthesia. Our findings demonstrated that dexmedetomidine use reduced EA incidence in children receiving either sevoflurane or desflurane. A meta-analysis [29] indicated that compared with placebo use, dexmedetomidine use reduced sevoflurane-induced EA incidence in children. Nevertheless, the differences in EA incidence between children receiving sevoflurane and desflurane remained unclear [30, 31]. Lim et al. [30] reported that the EA incidence between children receiving sevoflurane and desflurane was comparable, whereas He et al. [31] reported that the effects of sevoflurane use were considerably superior to those of desflurane use. Despite the nonsignificant differences observed in EA incidence, the meta-analysis of Lim et al. [30] indicated higher EA risk in the desflurane group (31%) than in the sevoflurane group (25%). The reasons for this discrepancy included the differences in the numbers of included studies and the participants undergoing different types of surgery, which involved different premedication types and the use of different tools to assess EA incidence. Moreover, in a recent study on desflurane, halothane, isoflurane, propofol, and sevoflurane in pediatric anesthesia, desflurane led to the highest EA incidence [32]. Our results indicated that dexmedetomidine use considerably decreased the EA incidence when patient received sevoflurane or desflurane. Besides, our current subgroup analysis results showed that dexmedetomidine use reduced more EA incidence in the sevoflurane group than in the desflurane group. Thus, the results indicated that dexmedetomidine potentially reduces EA incidence after the administration of a volatile anesthetic, particularly sevoflurane.

The reduction of EA incidence may depend on the administration route, timing, and duration of the intervention. EA risk was reduced only when propofol bolus was administered at the end of the sevoflurane anesthesia but not at anesthesia induction [33]. A previous meta-analysis [28] indicated that when dexmedetomidine was administered intravenously, it significantly reduced EA incidence. In contrast, when dexmedetomidine was administered orally, intranasally, and caudally, it did not reduce EA incidence. However, this finding was limited by only one trial for each administration route, except for the intravenous route. The results of another meta-analysis indicated that EA incidence decreased when dexmedetomidine was administered solely during the postoperative period [8]. Our findings indicate that dexmedetomidine administered as bolus infusion in the peri-induction period or at the end of surgery or as a continuous intraoperative infusion reduced EA incidence. Among the aforementioned administration strategies, continuous intraoperative infusion exhibited the largest effect size. The discrepancy between the results of previous studies and the current research may be due to differences in the operation time for various surgeries or dexmedetomidine pharmacokinetics. In our included studies, the operation time of strabismus surgery was relatively short (lasting around 15–48 minutes), whereas the terminal elimination half-life of intravenous dexmedetomidine was relatively long (lasting from 2.1 to 3.1 hours) [34]. This finding was derived from the few small-scale studies included in this meta-analysis. Therefore, additional high-quality large-scale studies assessing the optimal timing of dexmedetomidine administration are warranted.

PONV, which is a common phenomenon in patients who have undergone pediatric strabismus surgery, can potentially led to considerable dehydration, electrolyte imbalance, aspiration pneumonia risk, delayed hospital discharge, and an increased number of unscheduled hospital admissions, all of which increase healthcare costs [35, 36]. In their meta-analysis, Jin et al. [37] reported that compared with placebo, dexmedetomidine had a more significant prophylactic antiemetic effect in both adult and pediatric patients under general anesthesia; however, only one of the five included RCTs investigated children undergoing strabismus surgery [37]. Similarly, in the current meta-analysis, six studies indicated that compared with placebo use, dexmedetomidine use reduced PONV occurrence.

In our literature review, we found scant high-level evidence supporting premedication preventing PONV occurrence after pediatric strabismus surgery. In their 2014 meta-analysis, Shen et al. [38] reported that the prophylactic administration of dexamethasone, ondansetron, or their combination in pediatric strabismus surgery can reduce PONV incidence. In a 2016 guideline, ondansetron combined with dexamethasone—but not dexmedetomidine alone—was recommended to increase the effectiveness of PONV prevention in children scheduled for strabismus surgery [39]. In a 2019 meta-analysis [36], compared with placebo use, premedication with clonidine (another α2 agonist) decreased PONV incidence by 17% in children undergoing strabismus surgery, which is in agreement with our findings. Moreover, for PONV incidence, we found 0% heterogeneity and narrow CIs for its RR. Therefore, we have confidence in recommending dexmedetomidine use for PONV prevention in pediatric strabismus surgery. However, we could not compare dexmedetomidine with other commonly used antiemetics because few relevant head-to-head comparative studies were identified. An additional study clarifying this issue is warranted.

Postoperative pain is a major cause of morbidity related to pediatric anesthesia. Children undergoing strabismus surgery constitute a very-high-risk group for postoperative pain. Because elevated postoperative pain is associated with EA occurrence, a European Society for Pediatric Anesthesiology Pain Committee guideline [40] suggests postoperative pain management for six frequently performed procedures in children, but not including pediatric strabismus surgery. Dexmedetomidine has analgesic properties and has been widely used in various surgeries in adults. It can effectively relieve pain intensity, extend the pain-free period, and reduce opioid consumption during the postoperative recovery of adults under general anesthesia [41]. However, no suggestions have been provided in the literature on dexmedetomidine use for reducing postoperative pain in children [40]. Our current findings suggested that dexmedetomidine reduces the number of patients requiring analgesia and the pain intensity among children undergoing strabismus surgery. This result is consistent with those of previous studies [36]. Therefore, our results may be crucial for updating clinical practice guidelines in the future.

Regarding adverse events, we found that compared with placebo use, dexmedetomidine use reduced relative OCR risk by 37%. However, this finding should be interpreted with caution. Because OCR is associated with various triggering stimuli, the traction to the extraocular muscles was observed most commonly [42]. Moreover, compared with traction to other ocular muscles, traction to the medial rectus increased more OCR incidence [42]. However, in the included studies, this information was not clearly mentioned. Moreover, the heterogeneity was high and CI was wide in the included studies, which downgraded the certainty of the obtained evidence. Therefore, we suggest that anesthesiologists should consider using dexmedetomidine to reduce OCR occurrence in patients undergoing pediatric strabismus surgery. In addition, dexmedetomidine use as a preventative strategy may increase sedation and thus should be balanced against the risk of delaying PACU discharge [27]. We also found that dexmedetomidine does not cause a substantial difference in PACU stay duration. Consistent with our finding, Cho et al. [9] reported that dexmedetomidine does not influence the time to PACU discharge in children receiving tonsillectomy. By contrast, Pickard et al. [43] reported that α2 agonist use increases recovery time by <4 min, which is unlikely to be clinically relevant.

The current systematic review and meta-analysis has several limitations. First, the sample size of each included RCT was small. Moreover, all the studies were conducted at a single center and thus might have been subjected to small study effect biases. Second, dexmedetomidine administration routes and dosages varied significantly among the included studies; thus, we cannot rule out their influence on our results. Third, insufficient evidence was available in the included studies for comparing the efficacy of dexmedetomidine with that of other active

comparators. However, a network meta-analysis indicated that the effects of dexmedetomidine combined with sevoflurane were superior to those of ketamine, propofol, fentanyl, midazolam, sufentanil, remifentanil, and clonidine in reducing EA risk in children undergoing ophthalmic surgery [44]. Fourth, the standard postoperative management of PONV and postoperative pain varied among the included studies. Fifth, we found publication bias for EA incidence, which reduced the certainty of our current evidence.

## Conclusions

In the current meta-analysis, perioperative dexmedetomidine use was found to be associated with reduced EA incidence in patients who had undergone pediatric strabismus surgery, particularly those who received sevoflurane as the main anesthetic and dexmedetomidine as a continuous infusion. Moreover, perioperative dexmedetomidine use reduced PONV incidence, the number of patients requiring analgesia, and postoperative pain intensity. Finally, perioperative dexmedetomidine use may reduce OCR incidence but may not influence PACU stay duration.

## Supporting information

**S1 Checklist. PRISMA checklist.**
(DOC)

**S1 Table. Search strategy.**
(DOCX)

**S1 Fig. Forest plot for subgroup analyses of EA incidence based on the main anesthetic used.**
(TIF)

**S2 Fig. Forest plot for subgroup analyses of EA incidence based on the dexmedetomidine administration method.**
(TIF)

**S3 Fig. Forest plot for subgroup analyses of EA incidence based on the measurement method used.**
(TIF)

**S4 Fig. Sensitivity analysis results obtained when omitting one study at a time and calculating the pooled RRs for the remaining studies.**
(TIF)

**S5 Fig. Sensitivity analysis results by restricting studies at a low risk of bias.**
(TIF)

**S6 Fig.** (A) Funnel plots and (B) Egger's test results for EA incidence. The pseudo 95% CIs computed as part of the analyses were used to obtain the funnel plots and Egger's test results. The pseudo 95% CIs corresponded to the expected 95% CIs for a given standard error (SE).
(TIF)

**S7 Fig.** (A) Funnel plots and (B) Egger's test results for severe EA incidence. The pseudo 95% CIs computed in the analyses were used to obtain the funnel plots and Egger's test results. These CIs corresponded to the expected 95% CI for a given SE.
(TIF)

**S8 Fig.** (A) Funnel plots and (B) Egger's test results for PONV incidence. The pseudo 95% CIs computed in the analyses were used to obtain the funnel plot and Egger's test results. These CIs corresponded to the expected 95% CIs for a given SE.
(TIF)

**S9 Fig.** (A) Funnel plots and (B) Egger's test results for the number of patients requiring rescue analgesia. The pseudo 95% CIs computed in the analyses were used to obtain the funnel plots and Egger's test results. These CIs correspond to the expected 95% CIs for a given SE.
(TIF)

**S10 Fig.** (A) Funnel plots and (B) Egger's test results for pain scores. The pseudo 95% CIs computed in the analyses were used to obtain the funnel plots and Egger's test results. These CIs corresponded to the expected 95% CIs for a given SE.
(TIF)

**S11 Fig.** (A) Funnel plots and (B) Egger's test results for OCR incidence. The pseudo 95% CIs computed in the analyses were used to obtain the funnel plot and Egger's test results. These CIs corresponded to the expected 95% CI for a given SE.
(TIF)

**S12 Fig.** (A) Funnel plots and (B) Egger's test results for PACU stay duration. The pseudo 95% CIs computed in the analyses were used to obtain the funnel plots and Egger's test results. These CIs corresponded to the expected 95% CIs for a given SE.
(TIF)

**S13 Fig. Trial sequential analysis for EA incidence.** The risk of a type I error was maintained at 5% with 90% power. The variance was calculated from data obtained from the trials included in this meta-analysis. A clinically meaningful intervention effect for EA incidence was set to a 50% relative risk reduction based on an assumption of a 47% proportion of the control group. The result showed solid evidence indicating dexmedetomidine had a lower EA incidence compared to placebo.
(TIF)

**S14 Fig. Trial sequential analysis for severe EA incidence.** The risk of a type I error was maintained at 5% with 90% power. The variance was calculated from data obtained from the trials included in this meta-analysis. A clinically meaningful intervention effect for severe EA incidence was set to a 50% relative risk reduction based on an assumption of a 39% proportion of the control group. The result showed solid evidence indicating dexmedetomidine had a lower severe EA incidence compared to placebo.
(TIF)

**S15 Fig. Trial sequential analysis for PONV incidence.** The risk of a type I error was maintained at 5% with 90% power. The variance was calculated from data obtained from the trials included in this meta-analysis. A clinically meaningful intervention effect for PONV incidence was set to a 50% relative risk reduction based on an assumption of a 22.3% proportion of the control group. The result showed solid evidence indicating dexmedetomidine had a lower PONV incidence compared to placebo.
(TIF)

**S16 Fig. Trial sequential analysis for patients requiring rescue analgesia.** The risk of a type I error was maintained at 5% with 90% power. The variance was calculated from data obtained from the trials included in this meta-analysis. A clinically meaningful intervention effect for patients requiring rescue analgesia was set to a 50% relative risk reduction based on an

assumption of a 42% proportion of the control group. The result showed solid evidence indicating dexmedetomidine had a lower proportion of patients requiring rescue analgesia compared to placebo.
(TIF)

**S17 Fig. Trial sequential analysis for postoperative pain scores.** The risk of a type 1 error was maintained at 5% with a power of 90%. The variance was calculated from the data obtained from the included trials. A clinically significant anticipated mean difference in the postoperative pain scores was set to 1.73 based on the pooled result of our meta-analysis. The result showed solid evidence indicating dexmedetomidine had fewer postoperative pain scores compared to placebo.
(TIF)

**S18 Fig. Trial sequential analysis for OCR incidence.** The risk of a type I error was maintained at 5% with 90% power. The variance was calculated from data obtained from the trials included in this meta-analysis. A clinically meaningful intervention effect for OCR incidence was set to a 50% relative risk reduction based on an assumption of a 42.5% proportion of the control group. The result implied that more study needs to be conducted before the effect of dexmedetomidine on the reduction of OCR incidence can be definitively determined.
(TIF)

**S19 Fig. Trial sequential analysis for PACU stay duration.** The risk of a type 1 error was maintained at 5% with a power of 90%. The variance was calculated from the data obtained from the included trials. A clinically significant anticipated mean difference in expulsion times was set to 1.33 hours based on the pooled result of our meta-analysis. The result was inconclusive for PACU stay duration which did not differ between the dexmedetomidine and placebo groups.
(TIF)

## Acknowledgments

This manuscript was edited by Wallace Academic Editing.

## Author Contributions

**Conceptualization:** Fu-Wei Chiang, Jin-Lin Chang, Yuan-Pin Hsu.

**Data curation:** Fu-Wei Chiang, Jin-Lin Chang, Shih-Chang Hsu, Kuo-Yuan Hsu, Karen Chia-Wen Chu, Yuan-Pin Hsu.

**Formal analysis:** Fu-Wei Chiang, Jin-Lin Chang, Shih-Chang Hsu, Kuo-Yuan Hsu, Karen Chia-Wen Chu, Chyi-Huey Bai, Yuan-Pin Hsu.

**Funding acquisition:** Jin-Lin Chang, Yuan-Pin Hsu.

**Investigation:** Fu-Wei Chiang, Jin-Lin Chang, Shih-Chang Hsu, Kuo-Yuan Hsu, Karen Chia-Wen Chu, Yuan-Pin Hsu.

**Methodology:** Jin-Lin Chang, Shih-Chang Hsu, Chyi-Huey Bai, Chiehfeng Chen, Yuan-Pin Hsu.

**Project administration:** Fu-Wei Chiang, Yuan-Pin Hsu.

**Resources:** Chun-Jen Huang, Chin-Wang Hsu, Yuan-Pin Hsu.

**Software:** Kuo-Yuan Hsu, Yuan-Pin Hsu.

**Visualization:** Chun-Jen Huang.

**Writing – original draft:** Fu-Wei Chiang, Jin-Lin Chang.

**Writing – review & editing:** Fu-Wei Chiang, Yuan-Pin Hsu.

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
