## [Decision Letter · Decision Letter 0]

9 Sep 2020

PONE-D-20-14846

Dexmedetomidine use in pediatric strabismus surgery: A systematic review and meta-analysis

PLOS ONE

Dear Dr. Hsu,

Thank you for submitting your manuscript to PLOS ONE. After careful consideration, we feel that it has merit but does not fully meet PLOS ONE’s publication criteria as it currently stands. Therefore, we invite you to submit a revised version of the manuscript that addresses the points raised during the review process.

We look forward to receiving your revised manuscript.

Kind regards,

Laura Pasin

Academic Editor

PLOS ONE

Journal Requirements:

2. Please ensure you have included the full electronic search strategy for at least one database and uploaded it as an additional file.

Additional Editor Comments (if provided):

Subanalyses on low-risk of bias studies would add value to your results and a TSA analysis would allow the reader to assess if the data are convincing enough to prove the effect.

Please add them in your revised work.

Reviewers' comments:

Reviewer's Responses to Questions

**Comments to the Author**

1. Is the manuscript technically sound, and do the data support the conclusions?

Reviewer #1: Yes

Reviewer #2: Yes

2. Has the statistical analysis been performed appropriately and rigorously? 

Reviewer #1: Yes

Reviewer #2: Yes

3. Have the authors made all data underlying the findings in their manuscript fully available?

Reviewer #1: Yes

Reviewer #2: Yes

4. Is the manuscript presented in an intelligible fashion and written in standard English?

Reviewer #1: Yes

Reviewer #2: Yes

5. Review Comments to the Author

Reviewer #1: I have read with pleasure the manuscript entitled: "Dexmedetomidine use in pediatric strabismus surgery: A systematic review and metaanalysis". It is a well written and conducted meta analysis. I have only some minor remarks:

Most of the paragraph about study charachteristics repeat data reported in Table 1. They are unnecessary, please cut this paragraph removing duplication of data.

In methods state if subgroup analysis was pre-planned and it was a post-hoc analysis. Moreover, I do not find subgroup analysis with one or two studies appropriated. It seem more reasonable to state that there were not enough studies to perform a meta-analysis.

I suggest adding a post-hoc analysis with low risk of bias trials.

page 26 line 345 :"Our results indicated that compared with placebo use, perioperative use of dexmedetomidine in pediatric strabismus surgery might reduce the.. " Un-necessary repetition, I suggest to remove this sentence.

Reviewer #2: Chiang and coauthors performed a nice systematic review and meta-analysis investigating the use of dexmedetomidine in pediatric strabismus surgery. Within the nine included studies, they found a reduction in emergence agitation, postoperative nausea and vomiting, and postoperative pain in patients treated with dexmedetomidine.

The study was conducted in accordance with PRISMA guidelines and prospectively registered in the PROSPERO database.

All relevant studies appear to be included. The statistics appear to be well-conducted and complete.

The paper is presented in an intelligible fashion and well-written.

The authors should be complimented for their effort. I don't have any suggestion for modifications.

6. PLOS authors have the option to publish the peer review history of their article (what does this mean?). If published, this will include your full peer review and any attached files.

Reviewer #1: No

Reviewer #2: **Yes: **Pasquale Nardelli, MD

---

## [Author Response · Author response to Decision Letter 0]

18 Sep 2020

Dear Editors and Reviewers

We are very pleased to have been given the chance to revise and resubmit our manuscript to be considered further for publication in PLOS ONE. The comments of the editor and reviewers have been helpful in guiding the revision of our manuscript. We have attempted to make appropriate changes and have addressed the questions raised as follows: 

Editor Comments:

1. Subanalyses on low-risk of bias studies would add value to your results and a TSA analysis would allow the reader to assess if the data are convincing enough to prove the effect. Please add them in your revised work.

Response: Thank you for this helpful comment. We have added a sensitivity analysis on low-risk of bias studies. (page 10, 160-161; page 20, line 246-247) We have also performed a trial sequential analysis to allow the reader to assess if the data are convincing enough to prove the effect. (page 10-11, line 168-179; page 24-25, line 326-338; page 46-47, line 689-729)

Reviewer #1 report:

1. I have read with pleasure the manuscript entitled: "Dexmedetomidine use in pediatric strabismus surgery: A systematic review and meta-analysis". It is a well written and conducted meta-analysis. I have only some minor remarks: Most of the paragraph about study characteristics repeat data reported in Table 1. They are unnecessary, please cut this paragraph removing duplication of data.

Response: Thank you for your comments. We have cut this paragraph and removed the repeated data reported in Table 1. (page 12, line 196-197)

2. In methods state if subgroup analysis was pre-planned and it was a post-hoc analysis. Moreover, I do not find subgroup analysis with one or two studies appropriated. It seems more reasonable to state that there were not enough studies to perform a meta-analysis. 

Response: Thank you for your comments. We agree that in subgroup analysis, subgroup analysis data from only one study is inappropriate, but subgroup analysis data from ≥ two studies are still debate and may be reasonable. Therefore, we have removed the subgroup analysis with data from only one study. In this case, we reported it narratively. (page 18, line 229-232)

3. I suggest adding a post-hoc analysis with low risk of bias trials. 

Response: Thank you for your comments. We have added a sensitivity analysis on low-risk of bias studies. (page 10, 160-161; page 20, line 246-247)

4. page 26 line 345:"Our results indicated that compared with placebo use, perioperative use of dexmedetomidine in pediatric strabismus surgery might reduce the... " Un-necessary repetition, I suggest to remove this sentence.

Response: Thank you for your suggestion. We have removed this sentence. (page 26)

Reviewer #2 report:

1. Chiang and coauthors performed a nice systematic review and meta-analysis investigating the use of dexmedetomidine in pediatric strabismus surgery. Within the nine included studies, they found a reduction in emergence agitation, postoperative nausea and vomiting, and postoperative pain in patients treated with dexmedetomidine. The study was conducted in accordance with PRISMA guidelines and prospectively registered in the PROSPERO database. All relevant studies appear to be included. The statistics appear to be well-conducted and complete. The paper is presented in an intelligible fashion and well-written. The authors should be complimented for their effort. I don't have any suggestion for modifications.

Response: Thanks for taking the time in reviewing our work and your compliments.

Journal Requirements:

Response: Thank you for your kind reminders. We are sure that our manuscript meets PLOS ONE's style requirements, including those for file naming.

2. Please ensure you have included the full electronic search strategy for at least one database and uploaded it as an additional file.

Response: Thank you for your kind reminders. We have provided our electronic search strategy in supplementary documents (S1 Table). (Page 7, line 104; S1 Table)

---

## [Editor Report · Decision Letter 1]

29 Sep 2020

Dexmedetomidine use in pediatric strabismus surgery: A systematic review and meta-analysis

PONE-D-20-14846R1

Dear Dr. Hsu,

We’re pleased to inform you that your manuscript has been judged scientifically suitable for publication and will be formally accepted for publication once it meets all outstanding technical requirements.

Kind regards,

Laura Pasin

Academic Editor

PLOS ONE
---

## [Editor Report · Acceptance letter]

1 Oct 2020

PONE-D-20-14846R1 

Dexmedetomidine use in pediatric strabismus surgery: A systematic review and meta-analysis 

Dear Dr. Hsu:

I'm pleased to inform you that your manuscript has been deemed suitable for publication in PLOS ONE. Congratulations! Your manuscript is now with our production department. 

Kind regards, 

on behalf of

Dr. Laura Pasin 

Academic Editor

PLOS ONE